# Charting the native architecture of *Chlamydomonas* thylakoid membranes with single-molecule precision

**Wojciech Wietrzynski[1,2†], Miroslava Schaffer[2†], Dimitry Tegunov[3†], Sahradha Albert[2], Atsuko Kanazawa[4], Jürgen M Plitzko[2], Wolfgang Baumeister[2], Benjamin D Engel[1,2*]**

[1]Helmholtz Pioneer Campus, Helmholtz Zentrum München, Neuherberg, Germany; [2]Department of Molecular Structural Biology, Max Planck Institute of Biochemistry, Martinsried, Germany; [3]Department of Molecular Biology, Max Planck Institute for Biophysical Chemistry, Göttingen, Germany; [4]MSU-DOE Plant Research Lab, Michigan State University, East Lansing, United States

**Abstract** Thylakoid membranes scaffold an assortment of large protein complexes that work together to harness the energy of light. It has been a longstanding challenge to visualize how the intricate thylakoid network organizes these protein complexes to finely tune the photosynthetic reactions. Previously, we used in situ cryo-electron tomography to reveal the native architecture of thylakoid membranes (Engel et al., 2015). Here, we leverage technical advances to resolve the individual protein complexes within these membranes. Combined with a new method to visualize membrane surface topology, we map the molecular landscapes of thylakoid membranes inside green algae cells. Our tomograms provide insights into the molecular forces that drive thylakoid stacking and reveal that photosystems I and II are strictly segregated at the borders between appressed and non-appressed membrane domains. This new approach to charting thylakoid topology lays the foundation for dissecting photosynthetic regulation at the level of single protein complexes within the cell.

**\*For correspondence:**
ben.engel@helmholtz-muenchen.
de

[†]These authors contributed equally to this work

**Competing interests:** The authors declare that no competing interests exist.

## Introduction

Membranes orchestrate cellular life. In addition to compartmentalizing the cell into organelles, membranes can organize their embedded proteins into specialized domains, concentrating molecular partners together to drive biological processes (*Malinsky et al., 2013*; *Case et al., 2019*). Due to the labile and often transient nature of these membrane domains, it remains a challenge to study how individual protein complexes are organized within them.

The question of membrane domain organization is especially pertinent to thylakoids, sheet-like membrane-bound compartments that produce oxygen while converting light energy into biochemical energy, thereby sustaining most of the life on Earth. These light-dependent photosynthetic reactions are driven by the concerted actions of four large protein complexes within the thylakoid membrane. Photosystem II (PSII), cytochrome $b_6f$ (cyt$b_6f$), and photosystem I (PSI) form an electron transport chain that produces NADPH and pumps protons into the thylakoid lumen. ATP synthase then uses this electrochemical gradient across the membrane to generate ATP. In most photosynthetic eukaryotes, the thylakoid membranes are subdivided into appressed regions that face adjacent membranes within thylakoid stacks (called grana in higher plants) and non-appressed regions that freely face the stroma (*Austin and Staehelin, 2011*; *Nevo et al., 2012*; *Pribil et al., 2014*; *Engel et al., 2015*). In both plants and algae, PSII and PSI appear to be segregated to the appressed and non-appressed membranes, respectively (*Goodenough and Staehelin, 1971*;

*Wollman et al., 1980*; *Pribil et al., 2014*; *Flori et al., 2017*). This lateral heterogeneity is believed to coordinate photosynthesis by concentrating different reactions within specialized membrane domains, while the redistribution of light-harvesting complex II (LHCII) antennas between these domains may enable adaptation to changing environmental conditions (*Minagawa and Tokutsu, 2015*; *Nawrocki et al., 2016*).

Much of what is known about the molecular organization of thylakoids comes from freeze-fracture electron microscopy (and the related deep etch technique), which provides views of membrane-embedded protein complexes within the cell (*Heuser, 2011*). Combined with biochemical fractionation (*Andersson and Anderson, 1980*; *Albertsson, 2001*), these membrane panoramas helped describe the lateral heterogeneity of thylakoids (*Goodenough and Staehelin, 1971*; *Staehelin, 1976*; *Olive et al., 1979*; *Wollman et al., 1980*; *Staehelin and Arntzen, 1983*). However, the platinum replicas produced by this technique have limited resolution and only provide access to random fracture planes through the membranes. More recently, atomic force microscopy (AFM) has been used to map the macromolecular organization of thylakoids (*Sznee et al., 2011*; *Johnson et al., 2014*; *Wood et al., 2018*). AFM can very precisely measure topology and thus, can distinguish between each of the major photosynthetic complexes within a hydrated membrane. However, the membranes must first be removed from the cell, and only one membrane surface can be visualized at a time. As an alternative, cryo-electron tomography (cryo-ET) has been shown to resolve PSII complexes within hydrated thylakoids (*Daum et al., 2010*; *Kouřil et al., 2011*; *Levitan et al., 2019*). Although multiple overlapping membranes can be imaged with this technique, the thylakoids in these studies were isolated from the chloroplast in order to produce sufficiently thin samples.

Dissecting the interrelationship between membrane domains and thylakoid architecture requires a molecular view of intact thylakoid networks within native cells. In pursuit of this goal, we combined cryo-focused ion beam milling (*Marko et al., 2007*; *Schaffer et al., 2017*) with cryo-ET (*Asano et al., 2016*) to image thylakoid membranes within vitreously-frozen *Chlamydomonas reinhardtii* cells. Several years ago, we demonstrated how this approach can capture the undisturbed membrane architecture of this green alga, revealing an elaborate system of stacked thylakoids (*Engel et al., 2015*). However, due to the lower resolution of the CCD cameras used at that time, this investigation was limited to a description of membrane architecture, without visualizing the protein complexes embedded within these membranes. Here, we leverage advances in direct detector cameras and the contrast-enhancing Volta phase plate (*Danev et al., 2014*) to resolve the thylakoid-embedded complexes, enabling us to describe the molecular organization of thylakoids in situ, within their native cellular context.

## Results and discussion

Our tomograms show numerous densities corresponding to ribosomes and photosynthetic complexes decorating the appressed and non-appressed surfaces of the thylakoid network (*Figure 1A–B*, *Figure 1—figure supplement 1*, *Video 1*). Close visual inspection revealed the unmistakable shapes of ATP synthase and PSII protruding into the stroma and lumen, respectively (*Figure 1C–D*). In order to map these decorating densities into the native thylakoid architecture, we developed a visualization approach called a 'membranogram', where densities from the tomogram are projected onto the surface of a segmented membrane (*Figure 1E*). The result is a topological view of the membrane surface that resembles AFM data. However, unlike AFM, membranograms can display the topology of both sides of each membrane within the cellular volume. By dynamically growing and shrinking the segmentations, the membranograms allow us to track how densities change as they extend from the membrane surface and compare these densities to mapped in molecular models of different complexes (*Video 2*). This enabled the manual assignment of thylakoid-associated complexes by considering whether the densities projected into the stroma or thylakoid lumen (*Figure 2A*) and then comparing the densities to known structures of each complex (*Figure 2—figure supplement 1*). We used manually picked positions to generate subtomogram averages of PSII and ATP synthase, structurally confirming that membranograms can correctly identify these complexes (*Figure 2—figure supplement 2*).

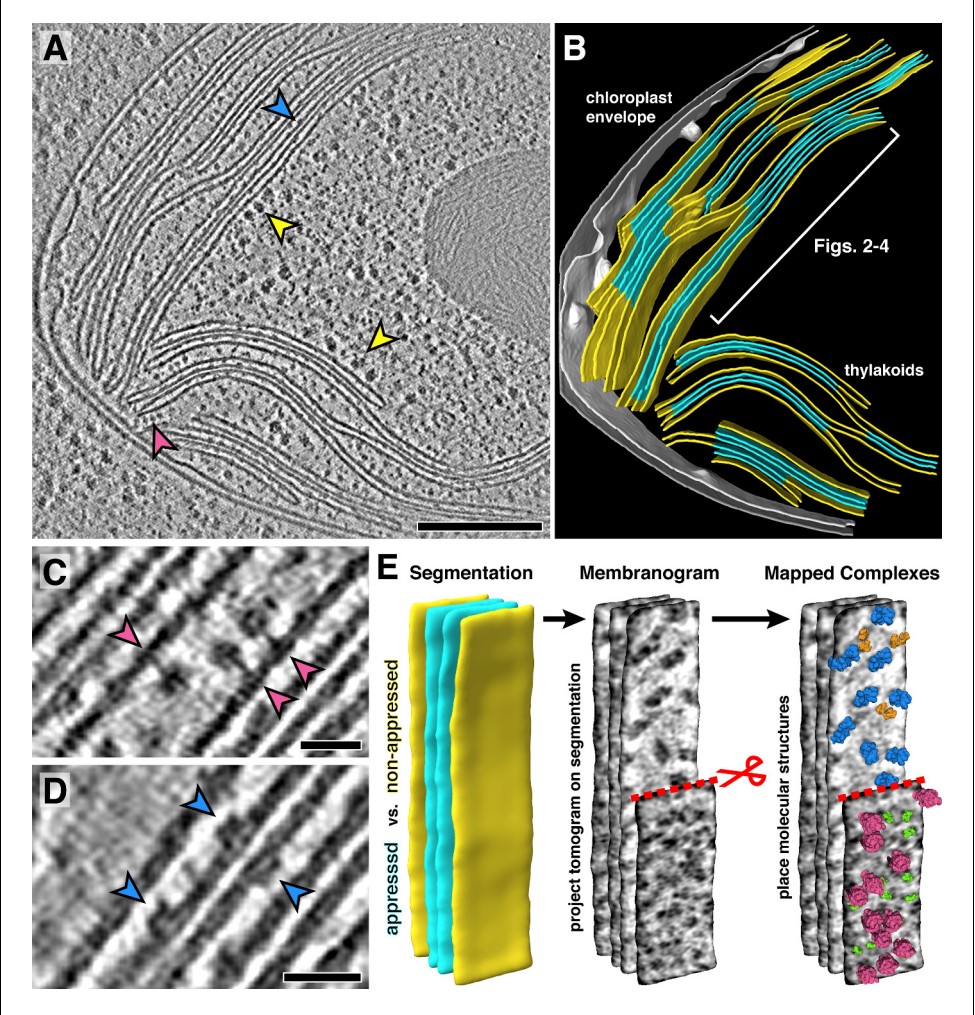

**Figure 1.** In situ cryo-electron tomography reveals the native molecular architecture of thylakoid membranes. (**A**) Slice through a tomogram of the chloroplast within an intact *Chlamydomonas* cell. Arrowheads point to membrane-bound ribosomes (yellow), ATP synthase (magenta), and PSII (blue). (**B**) Corresponding 3D segmentation of the chloroplast volume, with non-appressed stroma-facing membranes in yellow, appressed stacked membranes in blue, and the chloroplast envelope in grey. The thylakoid region used in *Figures 2–4* is indicated. (**C-D**) Close-up views showing individual ATP synthase (**C**) and PSII (**D**) complexes. (**E**) Mapping photosynthetic complexes into thylakoids using membranograms. Segmented membranes are extracted from the tomogram (left). Tomogram voxel values are projected onto the segmented surfaces, showing densities that protrude from the membranes. Here, densities ~2 nm above the membrane surface are shown (red scissors and dashed line: part of the non-appressed membrane has been removed to reveal luminal densities on the appressed membrane). Protein complexes are mapped onto membranograms based on the shapes of the densities and whether they protrude into the stromal or luminal space (blue: PSII, orange: cyt$b_6f$, green: PSI, magenta: ATP synthase). Scale bars: 200 nm in **A**, 20 nm in **C** and **D**. See *Videos 1* and *2*.

The online version of this article includes the following figure supplement(s) for figure 1:

**Figure supplement 1.** Tomogram overviews.

**Figure supplement 2.** The *mat3-4* strain has a similar 77K fluorescence spectrum profile to wild-type cells.

We used membranograms to examine the molecular organization within appressed and non-appressed domains of the thylakoid network (*Figure 2*). The complexes in appressed membranes (M2 and M3 in *Figure 2B–D*) could be reliably assigned. The stromal surfaces of these membranes showed no large densities, whereas two clearly distinguished classes of densities were seen on luminal surfaces: large PSII dimers and smaller cyt$b_6f$ dimers that extend ~4 nm and ~3 nm into the

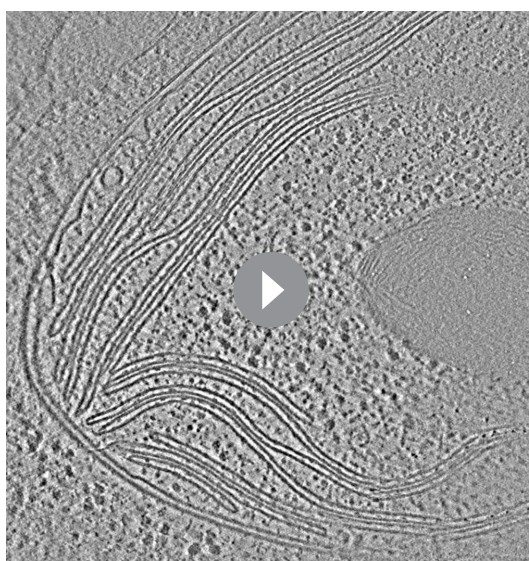

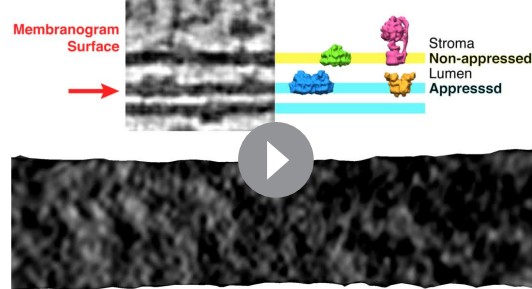

**Grow and Shrink Membrane Segmentation**

**Video 1.** In situ cryo-electron tomography reveals native thylakoid architecture with molecular clarity. Sequential slices back and forth in Z through the tomographic volume shown in *Figure 1A*. The yellow boxed region, focusing on the thylakoids shown in *Figures 2–4*, is then enlarged and displayed in the lower right corner of the video. Ribosomes and ATP synthase complexes can be seen bound to the stromal non-appressed thylakoid surfaces, while densities from PSII and cyt$b_6f$ extend from the appressed surfaces into the thylakoid lumen.

https://elifesciences.org/articles/53740#video1

**Video 2.** Mapping molecular complexes into thylakoid architecture with membranograms. Tomographic densities are projected onto the surface of a 3D membrane segmentation to produce a membranogram. The segmentation can be interactively grown and shrunk to visualize densities at different distances from the membrane. The bottom part of the video shows a membranogram of the luminal surface of an appressed thylakoid membrane. In the top part of the video, a moving red arrow indicates the position of the membranogram surface relative to the thylakoid architecture (shown both as real data and corresponding illustration). The video begins by growing and shrinking the segmentation, showing densities that begin within the appressed membrane and extend into the thylakoid lumen. Growing and shrinking the segmentation immediately adjacent to the membrane surface allows careful inspection of the thylakoid-bound densities. First, the positions and orientations (boxes with vector lines) of the PSII and cyt$b_6f$ (b6f) complexes are assigned. To-scale 3D structures of both complexes are then mapped onto the membranogram, showing good overlap with the tomographic densities.

https://elifesciences.org/articles/53740#video2

lumen, respectively. Non-appressed membranes (M1 in *Figure 2B–D*) were more complicated to analyze. We first assigned thylakoid-associated ribosomes and ATP synthases with high certainty by growing the segmentation ~10 nm away from the membrane. Assignment of PSI was more difficult: we marked single densities that protruded ~3 nm into the stroma and were not directly under a ribosome or ATP synthase. On the luminal side, almost no large densities were observed that could correspond to PSII. Although we did see frequent small dimers that resembled cyt$b_6f$, they were more difficult to identify due to increased background signal on the luminal surfaces of non-appressed membranes. We generated membrane models from the assigned particles (*Figure 2D*) to analyze how the complexes are arranged within the plane of the membrane. In total, we quantified 84 membrane regions from four tomograms (*Table 1*) and found clear evidence of lateral heterogeneity: PSII was almost exclusively found in the appressed regions, whereas PSI, ATP synthase, and ribosomes were restricted to the non-appressed regions. Cyt$b_6f$ was observed with almost equal abundance in both regions, with the caveat that we had lower confidence in our assignment of this complex in non-appressed membranes. Poly-ribosome chains were clearly resolved decorating the non-appressed thylakoids (*Figures 1A* and *2C–D*, *Figure 2—figure supplement 3*). In contrast, PSI, PSII, cyt$b_6f$, and ATP synthase were distributed relatively evenly along their respective membrane regions (nearest-neighbor distances plotted in *Figure 2—figure supplement 4*).

The concentrations we measured for different photosynthetic complexes were similar to previous biochemical estimates (*Allred and Staehelin, 1986*; *Vallon et al., 1991*; *Albertsson, 2001*) as well as AFM measurements (*Johnson et al., 2014*) and counts of 'membrane-embedded particles' from freeze-fracture (*Wollman et al., 1980*; *Staehelin, 2003*), indicating that direct comparisons can be

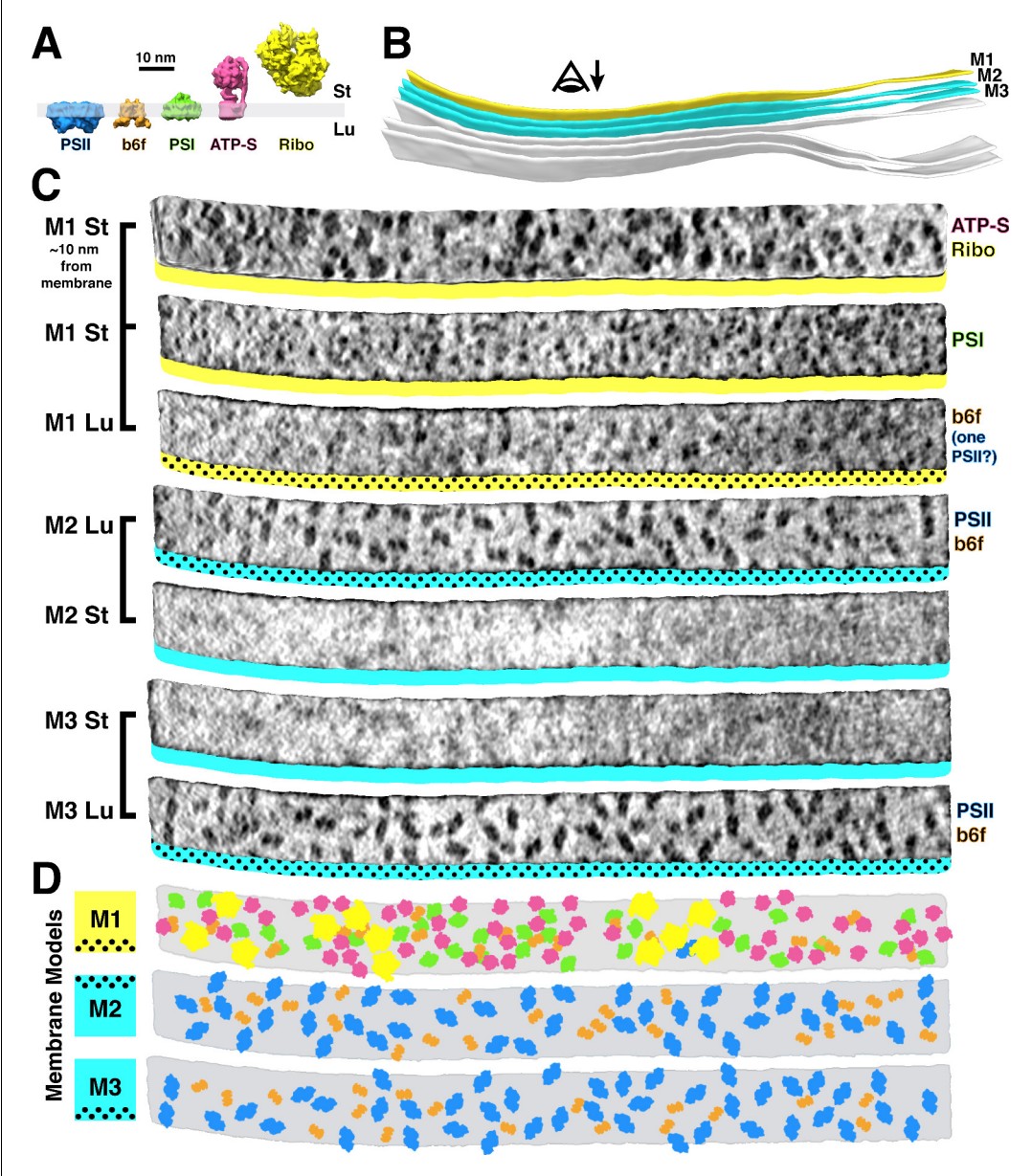

**Figure 2.** Mapping photosynthetic complexes within appressed and non-appressed thylakoid membranes. (**A**) Schematic of how each molecular complex extends from the membrane into the stroma (St) and thylakoid lumen (Lu). Into the lumen, large dimeric densities extend ~4 nm from PSII and small dimeric densities extend ~3 nm from cyt$b_6f$ (b6f). Into the stroma, a small monomeric density extends ~3 nm from PSI, the $F_1$ region of ATP synthase (ATP-S) extends ~15 nm, and thylakoid-bound ribosomes (Ribo) extend ~25 nm. (**B**) Three segmented thylakoids from the region indicated in *Figure 1B*. Membranes 1–3 (M1-M3, yellow: non-appressed, blue: appressed) are examined by membranograms. The eye symbol with arrow indicates the viewing direction for the membranograms. (**C**) Membranogram renderings of M1-M3. All membranograms show the densities ~2 nm above the membrane surface, except the top membranogram, which was grown to display densities ~10 nm into the stroma. Stromal surfaces are underlined with solid colors, whereas luminal surfaces are underlined with a dotted color pattern. The complexes identified in each surface are indicated on the right. (**D**) Model representation of M1-M3, showing the organization of all of the thylakoid complexes (colors correspond to the schematic in **A**). For a gallery of the different complexes visualized by membranograms, see *Figure 2—figure supplement 1*.

The online version of this article includes the following figure supplement(s) for figure 2:

**Figure supplement 1.** Membranogram particle gallery.

**Figure supplement 2.** Subtomogram averages of PSII and ATP synthase generated from particle positions assigned by membranograms.

**Figure supplement 3.** Thylakoid-bound poly-ribosome chains.

**Figure supplement 4.** Distributions of nearest-neighbor distances for PSII, cyt$b_6f$, ATP synthase, and membrane-associated ribosomes.

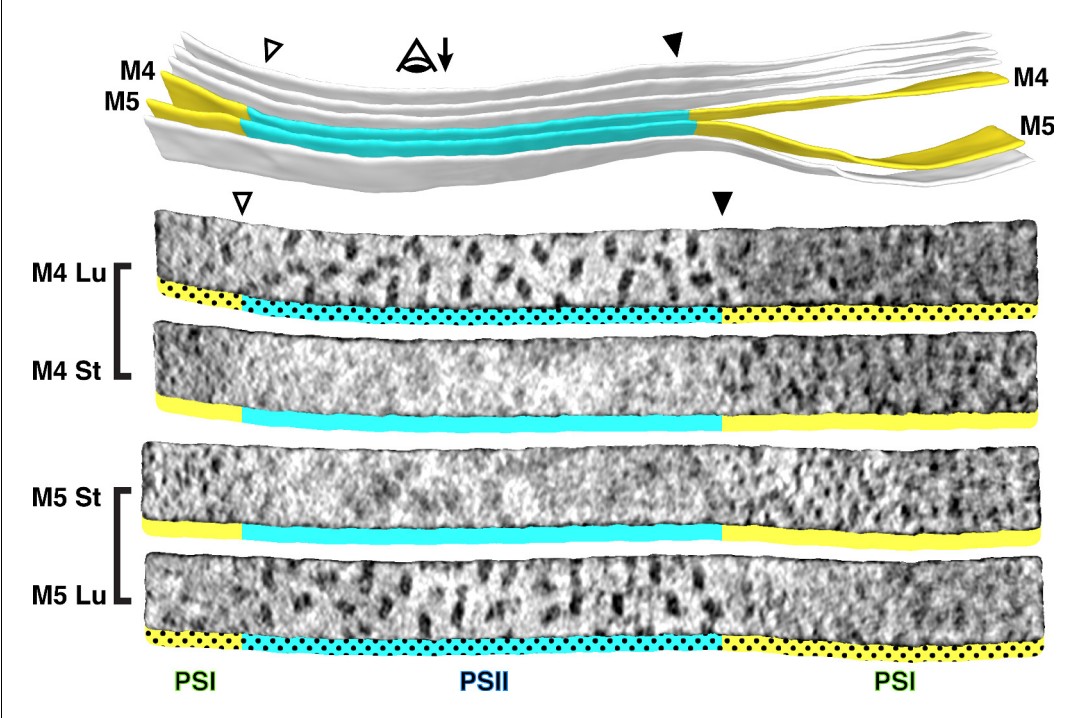

**Figure 3.** Strict segregation of PSII and PSI at transitions between appressed and non-appressed regions. Above: Three segmented thylakoids from the region indicated in *Figure 1B*. Membranes 4–5 (M4 and M5, yellow: non-appressed, blue: appressed) are examined by membranograms. The eye symbol with arrow indicates the viewing direction for the membranograms. Below: Membranograms of M4 and M5. All membranograms show the densities ~2 nm above the membrane surface. Stromal surfaces are underlined with solid colors, whereas luminal surfaces are underlined with a dotted color pattern. Transitions between appressed and non-appressed regions are marked with arrowheads. PSII is exclusively found in the appressed regions, whereas PSI is exclusively found in the non-appressed regions, with sharp partitioning at the transitions between regions. For an additional example of how lateral heterogeneity of PSII and PSI is coupled to membrane architecture, see *Figure 3—figure supplement 1*.

The online version of this article includes the following figure supplement(s) for figure 3:

**Figure supplement 1.** Additional example of the strict lateral heterogeneity between PSI and PSII at the transition between appressed and non-appressed membrane domains.

made between cryo-ET data and these earlier studies. In particular, the concentration of PSII dimers that we observed in appressed regions (1122 particles/$\mu m^2$) was in agreement with previous freeze-fracture measurements from *Chlamydomonas* (1250 particles/$\mu m^2$) (*Wollman et al., 1980*). We observed very few PSII complexes in the non-appressed membranes (*Figure 2—figure supplement 1B*), but it remains to be tested whether different conditions, such as damaging high light intensities, may increase the abundance of PSII outside the appressed domains. Our observation that cyt$b_6f$ is distributed relatively evenly between appressed and non-appressed membranes is also consistent with previous reports, although this ratio varies widely in the literature and may be modulated by physiological conditions (*Vallon et al., 1991*; *Staehelin, 2003*; *Kirchhoff et al., 2017*). The total number of densities that we counted on non-appressed membranes (4907 particles/$\mu m^2$) was similar to the number of particles previously observed by freeze-fracture in *Chlamydomonas* (4500 particles/$\mu m^2$) (*Wollman et al., 1980*). However, we are not aware of any previous direct measurement of PSI complexes to which we could compare. Spectroscopic estimates for the stoichiometry of PSII/PSI monomers in thylakoids vary widely from 0.8 to 2.1 (*Danielsson et al., 2004*; *Fan et al., 2007*); by cryo-ET, we measured a PSII/PSI monomer ratio of 2.1, but we note that PSI was not identified with high confidence (*Table 1*). The concentration of ATP synthase that we measured on non-appressed membranes (1652 particles/$\mu m^2$) was very similar to previous cryo-ET analysis of isolated spinach thylakoids (1770 particles/$\mu m^2$) (*Daum et al., 2010*) but more than twice the concentration initially proposed from freeze-fracture (710 particles/$\mu m^2$) (*Miller and Staehelin, 1976*; *Staehelin, 2003*).

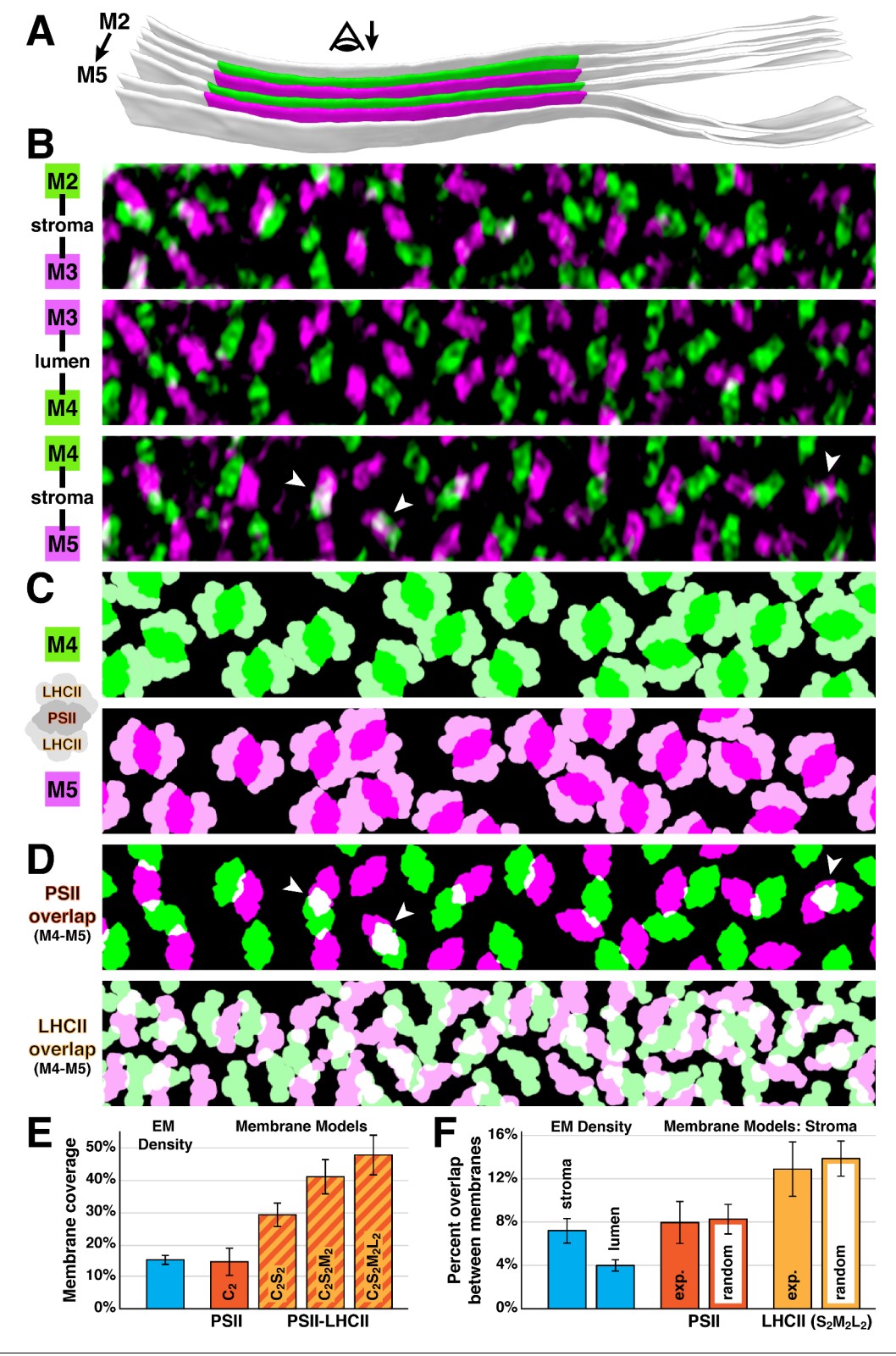

**Figure 4.** Native thylakoids can accommodate PSII-LHCII supercomplexes, which are randomly distributed across the stromal gap. (A) Three segmented thylakoids from the region indicated in *Figure 1B*. Membranes 2–5 (M2 to M5, alternating green and magenta) are examined by membranograms. The eye symbol with arrow indicates the viewing direction for the membranograms. (B) Overlays of two luminal surface membranograms, superimposed across the stromal gap (M2-M3, M4-M5) and thylakoid lumen (M3-M4). Membranograms are pseudocolored

*Figure 4 continued on next page*

Figure 4 continued

corresponding to **A**. Overlapping PSII complexes in the M4-M5 overlay are indicated with white arrowheads. (**C**) Model representations of M4 and M5, with $C_2S_2M_2L_2$-type PSII-LHCII supercomplexes positioned according to the luminal densities observed in the membranograms. The spacing indicates that appressed thylakoids can accommodate these supercomplexes. (**D**) M4-M5 overlays using the membrane models from **C**, separately showing the PSII core complexes (top) and LHCII antennas (bottom). Overlapping regions are white. White arrowheads correspond to the overlapping complexes in **B**. (**E**) Blue bar: the percentage of membrane surface area occupied by luminal EM density (15.0 ± 1.2%, N = 19 membranes) in membranograms (e.g., panel **B**). Red and red/yellow striped bars: the percentage of surface area in membrane models (e.g., panel **C**) occupied by $C_2$-type PSII core complexes (14.6 ± 4.1%) and $C_2S_2$-type (29.1 ± 3.7%), $C_2S_2M_2$-type (40.7 ± 5.2%), and $C_2S_2M_2L_2$-type (47.3 ± 6.0%) PSII-LHCII supercomplexes (N = 9 membranes). (**F**) Blue bars: the percentage of luminal EM density in membranograms that overlaps between adjacent appressed membranes spanning the stromal gap (7.2 ± 1.1%, N = 11 overlays) and thylakoid lumen (4.0 ± 0.5%, N = 6 overlays). See *Figure 4—figure supplement 1* for how the EM density was thresholded to calculate surface area and overlap. Red and yellow bars: using membrane models with $C_2S_2M_2L_2$-type PSII-LHCII supercomplexes, the percentage of PSII and LHCII surface area that overlaps between adjacent appressed membranes spanning the stromal gap (N = 4 overlays). The experimental measurements (exp.; PSII: 8.0 ± 1.9%, LHCII: 12.9 ± 2.5%) and simulations of complexes randomly positioned within the membrane (random; PSII: 8.3 ± 1.4%, LHCII: 13.8 ± 1.6%, N = 100 simulations per overlay) were not significantly different (p>0.05 from t-tests with Welch's correction for unequal variances: p=0.512 for PSII, p=0.762 for LHCII). See *Figure 4—figure supplement 2* and Materials and methods for how the random simulations were performed. Error bars denote standard deviation.

The online version of this article includes the following figure supplement(s) for figure 4:

**Figure supplement 1.** Calculations of membrane coverage and intermembrane overlap from EM density.

**Figure supplement 2.** Generation of random membrane models.

At the boundary between the appressed and non-appressed membranes of higher plants (i.e., the grana and stroma lamellae), it is widely believed that there is a specialized domain called the 'grana margin' where PSII and PSI intermix (*Anderson, 1989*; *Wollenberger et al., 1995*; *Kitmitto et al., 1997*; *Albertsson, 2001*). Although green algae do not have traditional grana, we were curious if such margin regions could be observed in *Chlamydomonas*. Using membranograms, we visualized the molecular organization of native thylakoids that transitioned between appressed and non-appressed regions (*Figure 3*, *Figure 3—figure supplement 1*). Strikingly, we observed a sharp boundary between PSII and PSI that exactly matched the division in thylakoid architecture. The photosystems did not intermix, and we saw no clear difference in particle abundance near the boundary. We therefore find no evidence for 'grana margins' in *Chlamydomonas* cells grown under these conditions (mixotrophic growth in moderate light, see Materials and methods).

What drives the strict lateral heterogeneity that we observe between appressed and non-appressed domains? PSI is presumably excluded from appressed membranes because its ~3 nm stromal density is too bulky to fit into the ~3 nm space between stacked thylakoids (*Daum et al., 2010*; *Kirchhoff et al., 2011*; *Engel et al., 2015*). Conversely, PSII and its associated LHCII antennas may induce thylakoid stacking, a causal relationship that would precisely limit PSII to appressed membranes. Several studies have observed semi-crystalline arrays of $C_2S_2$-type (*Boekema et al., 2000*; *Daum et al., 2010*) or $C_2S_2M_2$-type (*Kouřil et al., 2012*) PSII-LHCII supercomplexes in thylakoids isolated from higher plants, and it has been proposed that the overlap of LHCII or PSII between membranes mediates thylakoid stacking (*McDonnel and Staehelin, 1980*; *Boekema et al., 2000*; *Standfuss et al., 2005*; *Daum et al., 2010*; *Albanese et al., 2017*; *Albanese et al., 2020*). Although we observed randomly oriented PSII complexes instead of ordered arrays, we nonetheless looked for evidence of supercomplex interactions across native thylakoid stacks (*Figure 4*). We first

**Table 1.** Average concentrations of macromolecular complexes in native *Chlamydomonas* thylakoid membranes.

**Concentration of complexes**
[average number of complexes per square micrometer]

| | PSII | Cyt$b_6f$ | PSI | ATP-S | Ribo | Unknown | Total |
|---|---|---|---|---|---|---|---|
| Non-appressed | 24 | 501* | 1049* | 1652 | 113 | 1568 | 4907 |
| Appressed | 1122 | 631 | 2 | 0 | 0 | 170 | 1925 |

N = 84 membrane regions (51 non-appressed and 33 appressed) from four tomograms. 'Unknown' densities are particles that could not be assigned an identity, including particles found near the edges of the segmented membrane regions. Asterisks show classes of complexes that were identified with lower confidence. PSII densities are dimers, and thus the monomeric stoichiometry of PSII/PSI is 2.14. Densities that could correspond to free PSII monomers were seldom observed in membranograms and thus were not assigned.

created membranogram overlays of adjacent membranes spanning either the thylakoid lumen or stromal gap (*Figure 4B*). Then we generated membrane models by using the positions and rotational orientations of PSII luminal densities seen in the membranograms to place structures of $C_2S_2M_2L_2$-type PSII-LHCII supercomplexes (*Burton-Smith et al., 2019*; *Shen et al., 2019*; *Sheng et al., 2019*), the largest supercomplexes that have been isolated from *Chlamydomonas* (*Figure 4C–D*). Note that because LHCII barely protrudes from the membrane surface (*McDonnel and Staehelin, 1980*; *Standfuss et al., 2005*; *Johnson et al., 2014*) and thus is not well resolved in membranograms, we relied solely on the orientations of the PSII core complexes to place the supercomplex models. While the majority of $C_2S_2M_2L_2$-type supercomplexes fit within the plane of the membrane, we observed a ~3% in-plane overlap between the models (*Figure 4C*), indicating that some PSII may form smaller supercomplexes under the moderate light conditions that we examined (~90 µmol photons $m^{-2}s^{-1}$). It should be noted that the previously characterized $C_2S_2M_2L_2$-type supercomplexes were isolated from cells grown under lower light (20–50 µmol photons $m^{-2}s^{-1}$), which should favor larger supercomplex assemblies. Nevertheless, we observed that there is ample space within the appressed regions of *Chlamydomonas* thylakoids to accommodate large PSII-LHCII supercomplexes. Mapping in $C_2S_2M_2$-type supercomplexes, a slightly smaller arrangement that has been purified from higher plants (*Su et al., 2017*; *van Bezouwen et al., 2017*), resulted in almost no in-plane overlap between the models. Supercomplex models of different sizes occupied 47.3 ± 6.0% ($C_2S_2M_2L_2$), 40.7 ± 5.2% ($C_2S_2M_2$), and 29.1 ± 3.7% ($C_2S_2$) of the membrane surface area (*Figure 4E*), with cyt$b_6f$ occupying an additional 5.8 ± 1.6%. Thylakoids are ~70% protein (*Kirchhoff et al., 2002*), suggesting that other complexes such as extra LHCII antennas may occupy up to 20% of the surface area. This spacing should also allow room for rapid diffusion of plastoquinone between PSII and cyt$b_6f$ within appressed membranes.

The membranogram overlays showed very little overlap of PSII between adjacent membranes. In particular, PSII complexes were almost completely interdigitated across the thylakoid lumen, with only ~4% overlap of EM density (*Figure 4B and F*, *Figure 4—figure supplement 1*). PSII interdigitation has previously been predicted from spinach thylakoids with a contracted ~4.5 nm lumen, which sterically could only permit interdigitation (*Daum et al., 2010*). However, the thylakoids in our light-adapted *Chlamydomonas* cells have a ~9 nm lumen, which is enough space to permit face-to-face interactions of PSII across the lumen, yet the PSII complexes remain interdigitated. It has been suggested that interdigitation could facilitate diffusion of the soluble electron carrier plastocyanin through the lumen, while also enabling the lumen to contract and block diffusion in response to changing light conditions (*Kirchhoff et al., 2011*). To investigate interactions across the stromal gap, we measured the intermembrane overlap from the membrane models of $C_2S_2M_2L_2$-type supercomplexes and compared it to simulated data where the same number of supercomplexes were randomly positioned (*Figure 4D and F*, *Figure 4—figure supplement 2*). We found that the overlap of PSII-to-PSII and LHCII-to-LHCII were both statistically indistinguishable from random. It remains to be tested whether randomly distributed interactions covering ~8% of the PSII surface area and ~13% of the $S_2M_2L_2$ LHCII surface area (disregarding the ~3% in-plane overlap) can mediate the tight adhesion between stacked thylakoid membranes.

In situ cryo-ET has enabled us to visualize the native architecture of thylakoids with molecular resolution, revealing several striking findings that raise questions about how thylakoids partition their proteins and form appressed stacks. We observe sharp boundaries between PSII and PSI that are tightly coupled to membrane architecture, with no evidence of intermixing at transitions between appressed and non-appressed membrane domains. This argues against the presence of 'grana margins'. However, more definitive conclusions will require investigation of the true grana stacks found in higher plants. Semi-crystalline arrays of PSII have been reported in thylakoids isolated from spinach, pea, and diatoms (*Staehelin, 1976*; *Boekema et al., 2000*; *Daum et al., 2010*; *Sznee et al., 2011*; *Goral et al., 2012*; *Kouřil et al., 2012*; *Levitan et al., 2019*). However, randomly arranged PSII has been observed even more frequently (*Goodenough and Staehelin, 1971*; *Staehelin, 1976*; *Olive et al., 1979*; *Wollman et al., 1980*; *Daum et al., 2010*; *Kouřil et al., 2011*; *Goral et al., 2012*; *Johnson et al., 2014*; *Wood et al., 2018*; *Levitan et al., 2019*), and thus the physiological relevance of PSII arrays remains debated (*Tsvetkova et al., 1995*; *Ruban and Johnson, 2015*; *Tietz et al., 2015*; *Charuvi et al., 2016*). In our first look at PSII organization within native membranes, we found that PSII complexes are randomly oriented within the appressed thylakoids of *Chlamydomonas*, with ample spacing to accommodate $C_2S_2M_2$-type and $C_2S_2M_2L_2$-type

supercomplexes. This observation supports the physiological relevance of these larger supercomplexes, which have been primarily characterized in vitro (*Caffarri et al., 2009*; *Su et al., 2017*; *van Bezouwen et al., 2017*; *Burton-Smith et al., 2019*; *Shen et al., 2019*; *Sheng et al., 2019*). However, it remains to be proven whether semi-crystalline PSII arrays form inside the native cellular environment or whether they are a consequence of isolating thylakoid membranes. Unlike earlier techniques, our combination of in situ cryo-ET with membranograms has enabled us to visualize the organization of PSII across multiple adjacent stacked thylakoids within native cells. We determined that PSII supercomplexes randomly overlap between appressed membranes, challenging the idea that thylakoid stacking is mediated by specific static interactions between LHCII or PSII proteins across the stromal gap. Beyond these observations, our study establishes the methodological foundation to explore the conservation of thylakoid molecular architecture with other ecologically important phototrophs and dissect the mechanisms that adapt this architecture to changing environmental conditions.

## Materials and methods

### Cell culture

We used the *Chlamydomonas reinhardtii* strains *mat3-4* (CC-3994) (*Umen, 2001*) and wild-type CC-125, provided by the *Chlamydomonas* Resource Center, University of Minnesota, MN. The *mat3-4* strain has smaller cells that vitrify better by plunge freezing. Comparative 77K fluorescence measurements showed that *mat3-4* and wild-type cells had similar spectra under the conditions that the cells were grown and frozen onto EM grids (*Figure 1—figure supplement 2*). Thus, our description of the arrangement of photosynthetic complexes within *mat3-4* thylakoids is likely comparable to wild-type thylakoids.

For cryo-ET, *mat3-4* cells were grown to mid-log phase (1000–2000 cells/μL) in Tris-acetate-phosphate (TAP) medium under constant light conditions (~90 μmol photons $m^{-2}s^{-1}$) and bubbling with normal air. For 77K measurements, both *mat3-4* and wild-type strains were grown under these same conditions, and measurements were made both immediately and after allowing cultures to sit in low light for 30 min.

### 77K measurements

Fresh colonies of CC-125 and CC-3944 were picked from agar plates and suspended in sterile air-bubbled flasks containing 50 mL TAP medium. The cultures were grown at room temperature under 90 μmol photons $m^{-2}s^{-1}$ light until reaching 1,000 cells/μL. Samples containing 15 mL culture suspension were transferred into duplicate 15 mL Falcon tubes. From one of these tubes, five 1 mL aliquots were transferred to Eppendorf tubes and frozen immediately in liquid nitrogen. The second Falcon tube was kept under dim light of ~10 μmol photons $m^{-2}s^{-1}$ for 30 min, with gentle mixing by inversion, and five 1 mL samples were then frozen as above. Fluorescence emission spectra were acquired as previously described (*Kanazawa et al., 2014*) under liquid nitrogen, using a custom-made apparatus. Excitation was from a 440 nm diode laser, the output of which was sent through one branch of a bifurcated optical fiber (diameter of 1 mm) to the sample. The emission light was collected through the second fiber and measured with a spectrometer (Ocean Optics HR200 + ER). The average 77K profile for each condition was composed of three independent biological replicates, each with five technical replicates (15 measurements total per condition).

### Vitrification and Cryo-FIB milling

Using a Vitrobot Mark 4 (FEI, Thermo Fisher Scientific), 4 μL of *mat3-4* cell culture was blotted onto R2/1 carbon-coated 200-mesh copper EM grids (Quantifoil Micro Tools) and plunge frozen in a liquid ethane/propane mixture. Grids were stored in liquid nitrogen until used for FIB milling. Cryo-FIB milling was performed following a previously reported procedure (*Schaffer et al., 2015*; *Schaffer et al., 2017*), using a Quanta dual-beam FIB/SEM instrument (FEI, Thermo Fisher Scientific), equipped with a Quorum PP3010 preparation chamber. Grids were clipped into Autogrid support rings modified with a cut-out on one side (FEI, Thermo Fisher Scientific). In the preparation chamber, the frozen grids were sputtered with a fine metallic platinum layer to make the sample conductive. Once loaded onto the dual-beam microscope's cryo-stage, the grids were

coated with a thicker layer of organometallic platinum using a gas injection system to protect the sample surface. After coating, the cells were milled with a gallium ion beam to produce ~100 nm-thick lamellas (*Figure 1* = 60–90 nm, *Figure 1—figure supplement 1A* = 100–140 nm, *Figure 1—figure supplement 1C* = 60–80 nm, *Figure 1—figure supplement 1E* = 110–140 nm). During transfer out of the microscope, the finished lamellas were coated with another fine layer of metallic platinum to prevent detrimental charging effects during Volta phase plate cryo-ET imaging, as previously described (*Mahamid et al., 2016*; *Schaffer et al., 2017*).

## Cryo-ET

After FIB milling, grids were transferred into a 300 kV Titan Krios microscope (FEI, Thermo Fisher Scientific), equipped with a Volta phase plate (*Danev et al., 2014*), a post-column energy filter (Quantum, Gatan), and a direct detector camera (K2 summit, Gatan). Prior to tilt-series acquisition, the phase plate was conditioned to about $0.5\pi$ phase shift (*Danev et al., 2017*). Using SerialEM software (*Mastronarde, 2005*), bidirectional tilt-series (separated at 0°) were acquired with 2° steps between −60° and +60°. Individual tilts were recorded in movie mode with 12 frames per second, at an object pixel size of 3.42 Å and a target defocus of −0.5 µm. The total accumulated dose for the tilt-series was kept below ~100 e-/Å$^2$. Each tomogram was acquired from a separate cell and thus is both a biological and technical replicate.

## Tomogram reconstruction

Frame alignment was performed with K2Align (https://github.com/dtegunov/k2align). Using IMOD software (*Kremer et al., 1996*), tilt-series were aligned with patch tracking, and bin4 reconstructions (13.68 Å pixel size) were created by weighted back projection. Of the 13 Volta phase plate tomograms acquired, four tomograms (*Figure 1*, *Figure 1—figure supplement 1*) were selected for analysis of photosynthetic complexes based on good IMOD tilt-series alignment scores and visual confirmation of well-resolved complexes at the thylakoid membranes.

## Membrane segmentation

Segmentation of chloroplast membranes was performed in Amira software (FEI, Thermo Fisher Scientific), aided by automated membrane detection from the TomoSegMemTV package (*Martinez-Sanchez et al., 2014*). Bin4 tomograms were processed with TomoSegMemTV to generate correlation volumes with high pixel intensity corresponding to membrane positions. The original bin4 tomograms and correlation volumes were imported into Amira, and the correlation volumes were segmented by 3D threshold-based selection, producing one-voxel-wide segmentations at the centers of the membranes. Using the 3D lasso selection tool, these segmentations were subdivided into appressed and non-appressed membrane regions (*Figure 1B*, *Figure 1—figure supplement 1B,D and F*). To generate membranograms visualizing complexes directly protruding from the membrane surface (PSII, PSI, cyt$b_6f$), the segmentations were grown by two voxels in all directions to produce a five-voxel-wide segmentation with a surface that matched the surface of the membrane. To generate membranograms visualizing complexes ~10 nm above the membrane surface (ATP synthase, membrane-bound ribosomes), segmentations were grown by 10 voxels in all directions. To produce smooth 3D surfaces, the segmented voxels were transformed into a polygonal mesh with the 'generate surface' command, decimated to 10% triangle density with the 'remesh surface' command, and smoothed with the 'smooth surface' command (50 iterations, 0.4 lambda). These surfaces were exported in the OBJ 3D model format.

## Membranograms and particle picking

Bin4 tomograms and corresponding membrane segmentations (OBJ models) were loaded into Membranorama software (https://github.com/dtegunov/membranorama; copy archived at https://github.com/elifesciences-publications/membranorama). This software projects tomographic density onto the surface of a 3D membrane segmentation to create a membranogram. Segmentations that had been grown by two voxels in Amira had surfaces that intersected densities protruding directly from the membrane surface (PSII, PSI, cyt$b_6f$). Segmentations that had been grown by 10 voxels in Amira had surfaces that intersected densities that were ~10 nm above the membrane surface (ATP synthase, membrane-bound ribosomes). Importantly, the Membranorama software can dynamically

grow and shrink 3D membrane segmentations in real time, enabling the user to interactively track how densities appear at different distances from the membrane surface (see *Video 2*). The software also allows to-scale 3D models of each molecular complex (PDB: 6IJJ for PSI, 6KAD for PSII, 1Q90 for cyt$b_6f$, 6FKF for ATP synthase, 5MMM for ribosome) (*Stroebel et al., 2003*; *Bieri et al., 2017*; *Hahn et al., 2018*; *Sheng et al., 2019*; *Su et al., 2019*) to be mapped onto the membrane and compared to the tomographic densities. This is accomplished interactively by clicking the surface of the 3D membrane segmentation and using the mouse wheel to rotate each particle in the plane of the membrane. Using these features, we manually assigned membrane-associated densities to different classes of macromolecular complexes based on their positions relative to the membrane and their characteristic structural features (*Figure 2—figure supplement 1*). For the luminal side of the membrane, we exclusively used segmentations that had been grown by two voxels. PSII was assigned to large dimeric densities projecting ~4 nm from the membrane surface, and cyt$b_6f$ was assigned to small dimeric densities projecting ~3 nm from the surface. For the stromal side of the membrane, we started with segmentations that had been grown by 10 voxels, assigning large round densities with ~25 nm diameters to ribosomes and smaller round densities with ~10 nm diameters to the $F_1$ subunit of ATP synthase. The stator of ATP synthase was often observed as a small density adjacent to the larger $F_1$ density (see *Figure 2—figure supplement 1B*). After assigning the positions of these two complexes, we next loaded the corresponding segmentation that had been grown by two voxels and assigned PSI to small round densities projecting ~3 nm from the surface that were not positioned directly under a ribosome or ATP synthase. This order of particle picking prevented misassignment of PSI to the stalk of ATP synthase or the translocon structures that attach ribosomes to thylakoid membranes. The clarity of membrane-associated densities varied between different membranes within the same tomogram, likely due to effects of the tomographic missing wedge on different membrane curvatures and orientations, as well as local differences in the quality of tilt-series alignment. Therefore, only larger complexes (ribosomes, ATP synthase, PSII) were assigned for membranes with lower-clarity densities. Of the 51 non-appressed membranes quantified in this study, all complexes were assigned in 28 membranes, all complexes except for cyt$b_6f$ were assigned in two membranes, only ATP synthase and ribosomes were assigned in four membranes, and only ribosomes were assigned in 17 membranes. All complexes were assigned in the 33 appressed membranes quantified in this study. Within appressed membranes, PSII and cyt$b_6f$ were assigned with high confidence. Within non-appressed membranes, ribosomes and ATP synthase were assigned with high confidence, whereas PSI and cyt$b_6f$ were assigned with lower confidence (denoted by an asterisk in *Table 1*).

## Subtomogram averaging

We used subtomogram averaging as a structural confirmation of the membranogram-picked positions for PSII and ATP synthase (*Figure 2—figure supplement 2*). Manually assigned positions and orientations were exported from Membranorama and used as starting parameters for real space subtomogram alignment in PyTom software (*Hrabe et al., 2012*). No classification was performed, and all membranogram-picked subvolumes were included in the averages (396 PSII particles, 639 ATP synthase particles).

## Analysis of protein complex organization

### Nearest-neighbor distances within the plane of a membrane

To measure nearest-neighbor distances between molecular complexes within the thylakoids (*Figure 2—figure supplement 4*), segmentations of essentially flat membrane regions were exported from Amira (FEI, Thermo Fisher Scientific) as MRC volumes. Coordinates of the particles (PSII, cyt$b_6f$, ATP synthase, ribosomes) assigned on each membrane region were exported from Membranorama software. Each membrane region was projected with its corresponding particles onto a flat plane to generate a 2D surface. Nearest-neighbor distances between the particles were then measured using Matlab scripts calculating the shortest path between objects. Clustered poly-ribosomes exclude large regions of the stromal surface (see *Figure 2C–D*, *Figure 2—figure supplement 3*), which can cause misleading nearest-neighbor measurements for ATP synthase. To avoid this, the membranes were cropped to exclude regions containing poly-ribosome clusters before measuring ATP synthase distances.

## Overlap between adjacent membranes using EM densities

To calculate overlap between EM densities from two adjacent membranes (*Figure 4F*), 11 membrane pairs separated by the stromal gap and six membrane pairs separated by the thylakoid lumen were segmented in Amira and imported into Membranorama software. Using the tools in Membranorama, surfaces from each membrane were selected and overlaid along a vector orthogonal to both membrane surfaces, ensuring a geometrically accurate superposition of the membrane densities. Images of the overlaid membranograms were then analyzed in Fiji software (*Schindelin et al., 2012*) as described in *Figure 4—figure supplement 1*. Thresholding and cropping the membranograms were necessary to decrease noise and avoid edge effects, respectively.

## Overlap between adjacent membranes using membrane models

LHCII light-harvesting antennas are poorly visualized by cryo-ET because they are almost entirely embedded within the thylakoid membrane. Nevertheless, we incorporated hypothetical PSII-associated LHCII complexes into our analysis by using the PSII core positions and orientations visualized in membranograms to generate membrane models containing $C_2S_2M_2L_2$-type PSII-LHCII supercomplexes. First, 3D segmentations of appressed membrane regions were exported from Amira (FEI, Thermo Fisher Scientific). In the Membranorama software, we manually aligned a correctly scaled 3D model of the $C_2S_2M_2L_2$-type PSII-LHCII supercomplex (PDB: 6KAD) (*Sheng et al., 2019*) with the EM densities observed for each PSII core particle. Coordinates and orientations of all the particles were exported from Membranorama. Using Matlab scripts, the PSII-LHCII supercomplex structure was filtered to 25 Å resolution and then mapped at the assigned positions and orientations into the 3D volume of the membrane segmentation, generating a 3D model of a one-voxel-thick membrane with embedded PSII-LHCII supercomplexes. To compare these experimentally determined models with simulated models containing randomly distributed supercomplexes, the same number of PSII-LHCII supercomplexes were placed one-by-one at random positions in the membrane (randomly selected voxels of the one-voxel-thick 3D membrane segmentation) and each time rotated by a random in-plane angle before placing the next supercomplex particle. Whenever a newly placed particle overlapped with a preexisting particle, rotation of the new particle was first attempted to avoid overlap, and if this failed, the particle was moved to a new random position (see *Figure 4—figure supplement 2* for a 2D schematic representation of this 3D procedure). 100 random models were generated for each appressed membrane region.

To measure the relative membrane area covered by the placed supercomplex structures (*Figure 4E*), the one-voxel-thick membrane segmentation was masked where it was intersected by the 3D structure volumes (PSII core alone, or PSII-LHCII supercomplexes of increasing size: $C_2S_2$-type, $C_2S_2M_2$-type, and $C_2S_2M_2L_2$-type). This masked 'occupied area' was then divided by the total number of voxels in the membrane segmentation to yield a percentage. To calculate the overlap of 'occupied area' between adjacent membranes (*Figure 4F*), the $C_2S_2M_2L_2$-type supercomplex structure was subdivided into separate PSII ($C_2$) and LHCII ($S_2M_2L_2$) regions using Chimera software (*Goddard et al., 2007*). Next, volumes of the complexes in one membrane were extended along a vector orthogonal to both membrane surfaces until they also intersected the adjacent membrane. These extended complex densities were then used to mask 'occupied area' on both membranes. Similar to the overlap calculation for EM density (detailed in *Figure 4—figure supplement 1*), the percent of overlap was calculated as the number of membrane voxels that were masked by complexes from both membranes (equivalent to white in *Figure 4—figure supplement 1*) divided by the sum of all masked voxels on the membrane (equivalent to white + green + magenta in *Figure 4—figure supplement 1*). 'White' overlap voxels masked by complexes from both membranes were only counted once in this calculation. The schematic models in *Figure 4C–D* provide a 2D representation of this 3D analysis.

## Acknowledgements

We thank Radostin Danev, Günter Pfeifer, and Matthias Pöge for technical and computational assistance, Stefan Pfeffer and Roberta Croce for helpful discussions, and Karin Engel for critically reading the manuscript. This work was supported by a grant from the Deutsche Forschungsgemeinschaft to BDE (EN 1194/1–1 as part of FOR 2092). AK was supported by the U.S. Department of Energy

(DOE), Office of Science, Basic Energy Sciences (BES) under Award number DE-FG02-91ER20021. Additional funding was provided by the Helmholtz Zentrum München and the Max Planck Society.

## Additional information

### Funding

| Funder | Grant reference number | Author |
|---|---|---|
| Deutsche Forschungsge-meinschaft | FOR 2092, EN 1194/1-1 | Benjamin D Engel |
| Basic Energy Sciences | DE-FG02-91ER20021 | Atsuko Kanazawa |
| Max-Planck-Gesellschaft | | Wolfgang Baumeister |
| Helmholtz-Gemeinschaft | | Benjamin D Engel |

The funders had no role in study design, data collection and interpretation, or the decision to submit the work for publication.

### Author contributions

Wojciech Wietrzynski, Data curation, Formal analysis, Validation, Investigation, Visualization, Methodology, Writing - original draft, Writing - review and editing; Miroslava Schaffer, Conceptualization, Data curation, Investigation, Methodology, Cryo-FIB milling and cryo-ET; Dimitry Tegunov, Conceptualization, Software, Visualization, Methodology, Writing - review and editing, Developed Membranorama software; Sahradha Albert, Formal analysis, Investigation, Visualization, Methodology; Atsuko Kanazawa, Data curation, Formal analysis, Investigation, 77K measurements; Jürgen M Plitzko, Conceptualization, Resources, Methodology; Wolfgang Baumeister, Conceptualization, Resources, Funding acquisition; Benjamin D Engel, Conceptualization, Data curation, Formal analysis, Supervision, Funding acquisition, Validation, Investigation, Visualization, Methodology, Writing - original draft, Project administration, Writing - review and editing

### Author ORCIDs

Wojciech Wietrzynski (iD) https://orcid.org/0000-0001-8898-2392
Dimitry Tegunov (iD) https://orcid.org/0000-0001-7019-3221
Atsuko Kanazawa (iD) https://orcid.org/0000-0001-9570-3419
Jürgen M Plitzko (iD) http://orcid.org/0000-0002-6402-8315
Benjamin D Engel (iD) https://orcid.org/0000-0002-0941-4387

### Decision letter and Author response

Decision letter https://doi.org/10.7554/eLife.53740.sa1
Author response https://doi.org/10.7554/eLife.53740.sa2

## Additional files

### Supplementary files

• Transparent reporting form

### Data availability

The four tomograms analyzed in this study have been deposited in the Electron Microscopy Data Bank (EMD-10780-10783). The membranorama software is freely available at: https://github.com/dtegunov/membranorama (copy archived at https://github.com/elifesciences-publications/membranorama).

The following datasets were generated:

| Author(s) | Year | Dataset title | Dataset URL | Database and Identifier |
|---|---|---|---|---|
| Schaffer M, Weitrzynski W, Engel BD | 2020 | In Situ Cryo-Electron Tomogram of the Chlamydomonas Chloroplast | https://www.ebi.ac.uk/pdbe/entry/emdb/EMD-10780 | Electron Microscopy Data Bank, EMD-10 780 |
| Schaffer M, Weitrzynski W, Engel BD | 2020 | In Situ Cryo-Electron Tomogram of the Chlamydomonas Chloroplast | https://www.ebi.ac.uk/pdbe/entry/emdb/EMD-10781 | Electron Microscopy Data Bank, EMD-10 781 |
| Schaffer M, Weitrzynski W, Engel BD | 2020 | In Situ Cryo-Electron Tomogram of the Chlamydomonas Chloroplast | https://www.ebi.ac.uk/pdbe/entry/emdb/EMD-10782 | Electron Microscopy Data Bank, EMD-10 782 |
| Schaffer M, Weitrzynski W, Engel BD | 2020 | In Situ Cryo-Electron Tomogram of the Chlamydomonas Chloroplast | https://www.ebi.ac.uk/pdbe/entry/emdb/EMD-10783 | Electron Microscopy Data Bank, EMD-10 783 |

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
