## [Decision Letter]

Thank you for submitting your article "Charting the native architecture of thylakoid membranes with single-molecule precision" for consideration by *eLife*. Your article has been reviewed by three peer reviewers, and the evaluation has been overseen by a Reviewing Editor and Vivek Malhotra as the Senior Editor. The following individuals involved in review of your submission have agreed to reveal their identity: Matt Johnson (Reviewer #1); Jun Minagawa (Reviewer #2); Andrew Staehelin (Reviewer #3).

The reviewers have discussed the reviews with one another and the Reviewing Editor has drafted this decision to help you prepare a revised submission.

Summary:

Understanding the molecular architecture of thylakoid membranes is a key component of photosynthesis research. Major advances in this field are often achieved with the introduction of technical advances. This research advance article uses focused ion beam milling and cryo electron tomography to examine the macromolecular organization of photosynthetic complexes within the thylakoid membrane of vitreously frozen *Chlamydomonas* cells. This advance leverages technology developments since the authors' previous work in *eLife* (Engel et al., 2015) and employs a custom software package to expand the previous study by mapping published protein structures to these membranes.

Their results reveal that photosystems I and II are segregated between appressed and non-appressed membrane regions, implying that their distribution is tightly controlled. Furthermore, the authors find that photosynthetic complexes are randomly organized within the membrane and that the distribution of PSII or LHCII within adjacent grana membranes suggests that membrane stacking is maintained by fluctuating and/or less defined interactions.

These findings address inconsistencies or disagreements in the literature. By placing high resolution structural information into cellular context, these results have implications for our understanding of the molecular mechanisms that organize photosynthetic machinery within thylakoid membranes.

Essential revisions:

1) The reviewers agree that the structures mapped to PSII-LHCII (5XNL of C2S2M2 with CP24 in pea as described in Su et al., 2017) and PSI-LHCI (5ZJI of PSI-LHCI-(phospho)LHCII supercomplex from maize state II in Pan et al., 2018) were surprising choices. Given that three cryo-EM structures of PSII-LHCII supercomplexes were recently reported in *C. reinhardtii* (Burton-Smith et al., 2019, Shen et al., 2019 and Sheng et al., 2019), and several PSI-LHCI models from *C. reinhardtii* are also available (Su et al., 2019; Kubota-Kawai (2019) JBC; Suga et al. (2019) Nat. Plants), the reviewers would like to see these more appropriate structures mapped to the tomograms using the membranorama software.

2) Remapping different structures to the membranograms would also mean revisiting the statistical analysis of distribution and overlap between these complexes across membranes.

---

## [Author Response]

Essential revisions:1) The reviewers agree that the structures mapped to PSII-LHCII (5XNL of C2S2M2 with CP24 in pea as described in Su et al., 2017) and PSI-LHCI (5ZJI of PSI-LHCI-(phospho)LHCII supercomplex from maize state II in Pan et al., 2018) were surprising choices. Given that three cryo-EM structures of PSII-LHCII supercomplexes were recently reported in *C. reinhardtii* (Burton-Smith et al., 2019, Shen et al., 2019 and Sheng et al., 2019), and several PSI-LHCI models from *C. reinhardtii* are also available (Su et al., 2019; Kubota-Kawai (2019) JBC; Suga et al. (2019) Nat. Plants), the reviewers would like to see these more appropriate structures mapped to the tomograms using the membranorama software.

We thank the reviewers for the suggestion. The reason that the initial draft of the manuscript used higher plant complexes was simply timing: the analysis in our study was finished before the 2019 *Chlamydomonas* structures were available. We agree that it is very appropriate to use the new *Chlamydomonas* structures in the final manuscript, so we have made new figures accordingly. Given the limited resolution that we used for mapped-in structures (25 Å), swapping the PSII and PSI core complexes made very little noticeable difference (e.g., see the new membrane models in Figure 2D). However, mapping in the newly-published structure of the C_2_S_2_M_2_L_2_ PSII-LHCII supercomplex (Sheng et al., 2019) produced a clear difference in the membranogram overlays (Figure 4C-D), and resulted in minor changes in the calculated membrane occupancy and inter-membrane overlap of the supercomplex-associated LHCII antennas. Please note that while C_2_S_2_M_2_L_2_ supercomplexes were mostly accommodated within the membranes, some in-plane overlap was introduced, unlike for C_2_S_2_M_2_. We discuss this in the text:

"Then we generated membrane models by using the positions and rotational orientations of PSII luminal densities seen in the membranograms to place structures of C_2_S_2_M_2_L_2_-type PSII-LHCII supercomplexes (Burton-Smith et al., 2019; Shen et al., 2019; Sheng et al., 2019), the largest supercomplexes that have been isolated from *Chlamydomonas* (Figure 4C-D). Supercomplex models of different sizes occupied 47.3% ± 6.0% (C_2_S_2_M_2_L_2_), 40.7 ± 5.2% (C_2_S_2_M_2_), and 29.1% ± 3.7% (C_2_S_2_) of the membrane surface area (Figure 4E), with cyt*b_6_f* occupying an additional 5.8 ± 1.6%."

Although we now base our primary analysis on C_2_S_2_M_2_L_2_ supercomplexes, we felt it was informative to provide the membrane occupancy for a variety of PSII models in our revised Figure 4E, including C_2_ (which is similar to the EM density overlap), C_2_S_2_, C_2_S_2_M_2_, and C_2_S_2_M_2_L_2_.

2) Remapping different structures to the membranograms would also mean revisiting the statistical analysis of distribution and overlap between these complexes across membranes.

After mapping in the new structures, we repeated the analysis of membrane occupancy and overlap of complexes between membranes (Figure 4E-F). As mentioned above, this resulted in minor changes in the numbers, with increased occupancy and LHCII overlap for C_2_S_2_M_2_L_2_ supercomplexes. However, there was no effect on the conclusion that PSII-LHCII supercomplexes are randomly arranged across stacked membranes (see Figure 4F).